# Research on Price Discovery in Financial Securities: Trends and Directions for Future Research

Prashant Sharma [1,*], Gaurav Agrawal [2], Geetika Arora [3], Dinesh Kumar Sharma [4] and Varun Chotia [5]

1   Jindal School of Banking and Finance (JSBF), O. P. Jindal Global University, Sonipat 131001, India
2   Atal Bihari Vajpayee-Indian Institute of Information Technology and Management, Gwalior 474015, India; gaurav@iiitm.ac.in
3   Business Administration, University of the People, Pasadena, CA 91101, USA; geetika.arora@uopeople.edu
4   School of Management, Gautam Buddha University, Greater Noida 201312, India; da.dinesh@gmail.com
5   Jaipuria Institute of Management Jaipur, Jaipur 302033, India; varun.chotia@jaipuria.ac.in
*   Correspondence: prashant.sharma@jgu.edu.in

**Abstract:** The futures contracts were introduced to act as hedging instruments and ensure the price discovery (referred to as PD hereafter) mechanism for the underlying securities. If the price movement of a futures contract leads the price movement of the underlying securities in the spot market, this confirms the existence of price discovery in the market. This study undertakes an in-depth analysis of past research in order to find research trends and directions for the future in the field of price discovery. The bibliometric analysis technique is used to analyse the existing literature. The study considers the 1431 documents collected from the Scopus database for the period of 1982–2021 to conduct the descriptive and network analysis of search results. The study identifies three key clusters, i.e., the foundation of the price discovery process (Cluster 1), the econometric tools and techniques to assess the price discovery process (Cluster 2), and price discovery under different market conditions and constraints (Cluster 3). After an in-depth content analysis of these clusters, the study provides suggestions for future research in the field of price discovery. The study is the first of its type to conduct an in-depth analysis of the literature of price discovery since inception, and provides directions for future research in the field.

**Keywords:** price discovery; bibliometric analysis; network analysis; VOS viewer; lead–lag relationship

## 1. Introduction

Futures contracts, as part of derivative products, are the financial instruments that allow a trader to enter a contract where the contract will be made for a future date while the price, quantity, quality, and delivery mechanism will be fixed at the time of entering the contract. The first derivatives exchange in the United States was established in 1848, followed by futures contracts on the Chicago Mercantile Exchange (CME) in 1972 and interest rate futures on the Chicago Board of Trade (CBT) in 1975 (Pauletto and Steve 2012). The two key objectives for the introduction of futures contracts were to act as a hedging instrument and ensure a price discovery mechanism for the underlying securities. The futures contracts enable traders to diversify their risk exposure with their current or future investments. These contracts are made for a future date, and the price of the same is available while entering into the contract. This indicates that the price of a future contract should reflect the expectations of different types of traders about the price of underlying securities in the future, and these contracts should serve as the important information criteria that derive the price of underlying securities in the spot market. If the price movement of a futures contract derives the price movement of the underlying securities in the spot market, this is considered the existence of price discovery (PD) in the market (Silber 1981).

Since the inception of these contracts, futures contracts on different securities, such as equities, currencies, commodities, interest rates, and cryptocurrencies, have been launched in different economies across the world. The trading volume of the derivatives market is significantly higher than that of the spot market, and as a result, the futures prices reflect the expectations of the investors more efficiently than the spot prices. This confirms that the price of a futures contract should lead to the price movement of the underlying security in the spot market, and this leads to the PD in the market.

Researchers have made significant attempts to assess the existence of PD among different asset classes. During the early phase of the existence of the PD discipline, a researcher tried to develop a framework to assess the PD mechanism. Hasbrouck (1995) introduced the "information sharing approach" and argued that for the same underlying security the prices should be similar in different markets where it is traded. Considering the sample of DOW 30 stocks, the study found the existence of PD in the US equity market. Gonzalo and Granger (1995) proposed a different econometric technique to assess PD. The authors suggested using the error correction mechanism when the variables have the same order of integration, that is, I(1). Fleming et al. (1996) studied the role of trading costs on the PD mechanism among the spot and derivatives markets and found that new information is reflected quickly among those securities that have low trading costs compared to those with high trading costs. The study also found that the cost of trading is lower with index-based securities than with individual stocks. Tse (1999) extended the literature by assessing the volatility spillover along with the PD mechanism in the US equity market. The study applied the bivariate EGARCH model to assess whether there is evidence of volatility spillover among the futures and spot indexes of the DJIA index. The results supported the bi-directional information flow, where the spillover from future-to-spot was higher than that from spot-to-future markets. Baillie et al. (2002) tried to develop the PD mechanism using the common factors of previous studies by Hasbrouck (1995) and Gonzalo and Granger (1995). The study found that these two popular PD models are related and lead to similar inferences when the error terms do not have problems with serial correlations. The subsequent studies by Hasbrouck (2003), Andersen et al. (2007), and Brogaard et al. (2014) on PD try to use the different data frequencies to assess the information sharing among the futures and spot markets. Hasbrouck (2003) used the intraday prices of the index futures and exchange-traded funds (ETFs) of the S&P 500 to assess the PD among the two markets and found the existence of PD among the futures and ETFs prices. Andersen et al. (2007) assessed the PD using the high-frequency data of three different markets, such as equities, debt, and forex, in three different economies: the USA, Germany, and Britain. Brogaard et al. (2014) assessed the PD mechanism among NASDAQ- and NYSE-listed firms by looking at the role of high-frequency traders. The study found that with the existence of high-frequency traders, the overall efficiency of information dissemination in pricing improves.

In the existing literature, numerous attempts have been made in the past to assess the existence of PD mechanisms in different markets. Most of these studies were empirical in nature and tried to assess whether price discovery exists in different asset classes across different economies. Some studies also tried to assess the role of volatility in the PD process among different underlying securities. Most of the existing literature is focused on the empirical examination of PD, while a detailed review of the discipline is still missing. The discipline has been in existence for the past four decades, and, to the best of our knowledge, no attempt has been made by the researchers to conduct an in-depth analysis of the existing literature on PD. Thus, it has become important to assess the past trends of the research publications on PD by conducting a comprehensive, in-depth analysis of the literature using the bibliometric method. These trends will be helpful for future researchers to select prominent authors and documents when they plan to contribute to the PD discipline. These trends will also enable the researchers to identify quality research work published in the domain, research trends, contributions by the most relevant researchers, contributions across geographies and institutions, and the connectedness among these aspects. In view of this, the aim of the present study tries to address the following research questions:

RQ1. What is the growth of publications on price discovery?

RQ2: Who are the most prominent authors and documents on price discovery?

RQ3: Who are the prominent contributors to price discovery?

RQ4. What are the collaboration patterns among the researchers on price discovery?

RQ5. What are existing and current research themes and topics on price discovery?

RQ6. What are the future research directions for research on price discovery?

## 2. Data and Methodology

This section provides a description of the sampling method, data collection, and analysis methods adopted to conduct the study.

### 2.1. Data and Sample

The bibliographic data collected from the Scopus Database indicates that the first paper on price discovery (PD) was published in 1982 and that a total of 1879 publications had been added to the body of knowledge on the topic by 2021 (refer to Figure 1). The investigation ultimately found a total of 1431 publications for this study after filtering the results to include only pertinent articles and reviews written in English. The study of these selected publications is provided in the following sections, considering several bibliometric characteristics associated with price discovery.

**Bibliometric Data Collection Strategy**

Database: Scopus

Search Key: "price discov\*" OR "discov\* of price"

Search Field: Title, Abstract, Keywords

Total Records: 1879

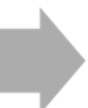

**Inclusion/Exclusions:**

Documents After Excluding Articles Published in 2022: 1867

Documents after Considering Articles Published in English Language: 1809

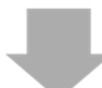

**Inclusion/Exclusions:**

Documents after Considering Articles Published in Economics, Econometrics and Finance and Business, Management and Accounting: 1523

Documents Published as Article and Review: 1431

Final Sample for Analysis: 1431

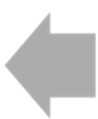

**Analysis Tools**

The descriptive analysis using Bibliometrix package (Aria & Cuccurullo, 2017)

Network analysis using VoSviewer (Van Eck and Waltman, 2010)

**Figure 1.** Description of sample selection and analysis method.

### 2.2. Bibliometric Methods and Their Use in Finance, Accounting and Related Fields

In recent years, there has been a significant increase in the use of bibliometric methods to conduct in-depth analyses of the literature in various disciplines, using a large pool of previously published studies. Pritchard (1969) defined bibliometrics as "the application of mathematics and statistical methods to books and other media of communication". Other researchers suggest the following: "The discipline of bibliometric methods has evolved

with the recommendations to use "bibliometric citations" (Broadus 1987), "co-citations" (Small 1973), "co-occurrences" (Koseoglu 2016), and "co-authorship" (Cheng et al. 2018). The researchers have used bibliometric methods to understand the intellectual structure and research trends in different fields (Podsakoff et al. 2008).

The application of bibliometric methods has become popular in recent years, and the business, finance, and accounting disciplines are no exceptions to this. Zheng et al. (2020) applied the scientometric methods in the construction industry to assess the research trends, while Ye et al. (2015) used social network analysis to understand the knowledge flow of the patient citation data.

The use of bibliometric methods in the financial and accounting disciplines has also increased significantly over the past few years, in the cases of green finance (Zhang et al. 2019), productivity in finance (Chung and Cox 1990), option pricing (Sharma et al. 2023), supply chain finance (Xu et al. 2018), Islamic banking and finance (Biancone et al. 2020), renewal energy finance (Elie et al. 2021), sustainable finance (Bui et al. 2020), and artificial intelligence and machine learning in finance (Goodell et al. 2021).

## 3. Results and Discussion

The results of the bibliometric analysis, including both descriptive analyses using the Bibliometrix package (Aria and Cuccurullo 2017) and network analysis (van Eck and Waltman 2010), are summarised and presented below.

### 3.1. Publication Productivity of Price Discovery Research (RQ1)

Out of the 1431 total publications examined in the field of price discovery, 1373 are conceptual and empirical papers and 58 are reviews. The first article in this field was "Price Trends at Livestock Auctions," written by Steven T. Buccola and published in 1982. From Table 1, in the first two decades (1982–2001) of price discovery research, only 10% of total papers were published, while in the first three decades (1982–2011), only 37% of papers were published. A total of 63% of the total publications were published in only one decade, i.e., the fourth decade (2012–2021). The recent five years (2017–2021) have produced 36% of the field's publications, and it has been extremely productive. The year 2021 saw 145 publications, which is the most ever. Overall, there has been an increasing trend in the growth of publications on price discovery research (refer to Figure 2).

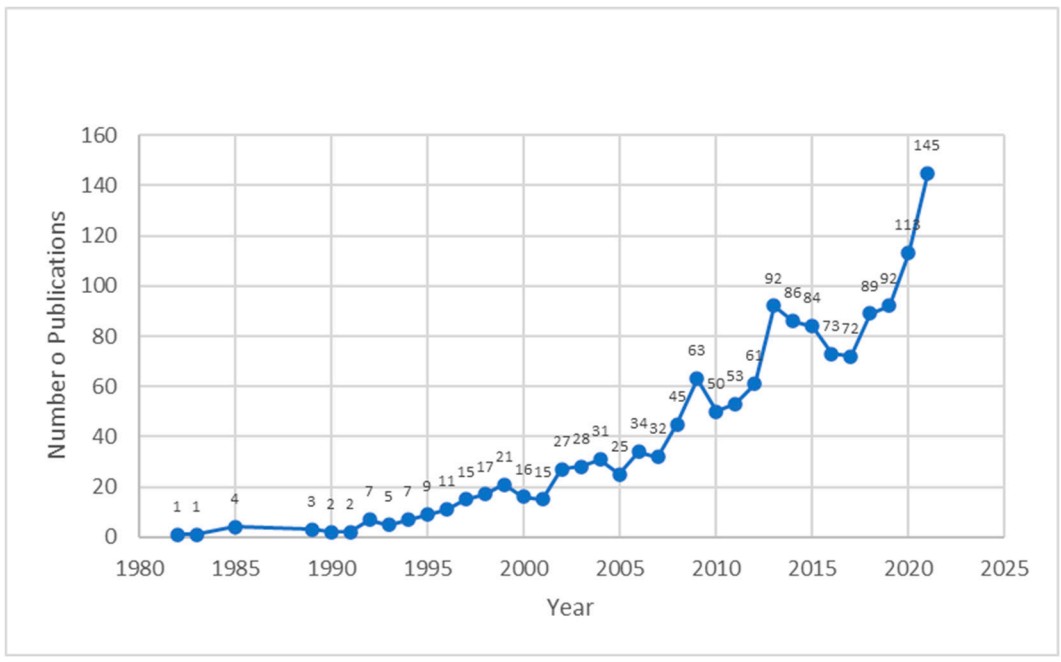

**Figure 2.** Article publications on PD across the years.

**Table 1.** Publication trends of PD research.

| Year | Number of Publications | Percentage | Cumulative Percentage | Total Citations |
|------|------------------------|------------|----------------------|-----------------|
| 1982 | 1 | 0.1% | 0.1% | 18 |
| 1983 | 1 | 0.1% | 0.2% | 9 |
| 1984 | 0 | 0.0% | 0.2% | 0 |
| 1985 | 4 | 0.3% | 0.4% | 19 |
| 1986 | 0 | 0.0% | 0.4% | 0 |
| 1987 | 0 | 0.0% | 0.4% | 0 |
| 1988 | 0 | 0.0% | 0.4% | 0 |
| 1989 | 3 | 0.2% | 0.7% | 84 |
| 1990 | 2 | 0.1% | 0.8% | 43 |
| 1991 | 2 | 0.1% | 0.9% | 128 |
| 1992 | 7 | 0.5% | 1.4% | 135 |
| 1993 | 5 | 0.3% | 1.8% | 87 |
| 1994 | 7 | 0.5% | 2.3% | 324 |
| 1995 | 9 | 0.6% | 2.9% | 1140 |
| 1996 | 11 | 0.8% | 3.7% | 529 |
| 1997 | 15 | 1.0% | 4.7% | 1025 |
| 1998 | 17 | 1.2% | 5.9% | 305 |
| 1999 | 21 | 1.5% | 7.4% | 905 |
| 2000 | 16 | 1.1% | 8.5% | 432 |
| 2001 | 15 | 1.0% | 9.5% | 1102 |
| 2002 | 27 | 1.9% | 11.4% | 1439 |
| 2003 | 28 | 2.0% | 13.4% | 2525 |
| 2004 | 31 | 2.2% | 15.5% | 1244 |
| 2005 | 25 | 1.7% | 17.3% | 1388 |
| 2006 | 34 | 2.4% | 19.7% | 940 |
| 2007 | 32 | 2.2% | 21.9% | 1299 |
| 2008 | 45 | 3.1% | 25.0% | 972 |
| 2009 | 63 | 4.4% | 29.5% | 1329 |
| 2010 | 50 | 3.5% | 32.9% | 1670 |
| 2011 | 53 | 3.7% | 36.6% | 1523 |
| 2012 | 61 | 4.3% | 40.9% | 1004 |
| 2013 | 92 | 6.4% | 47.3% | 1772 |
| 2014 | 86 | 6.0% | 53.3% | 1927 |
| 2015 | 84 | 5.9% | 59.2% | 1185 |
| 2016 | 73 | 5.1% | 64.3% | 613 |
| 2017 | 72 | 5.0% | 69.4% | 655 |
| 2018 | 89 | 6.2% | 75.6% | 845 |
| 2019 | 92 | 6.4% | 82.0% | 547 |
| 2020 | 113 | 7.9% | 89.9% | 414 |
| 2021 | 145 | 10.1% | 100.0% | 107 |

*3.2. Publication Impact of Price Discovery Research (RQ2)*

To understand the impact of published research, the citations received by the article are used as key criteria (Baker et al. 2020; Kumar et al. 2021). As per the study of global citations, most citations were received by J. Hasbrouck's (1995) article, "One Security, Many Markets: Determining the Contributions to Price Discovery," followed by Hendershott et al.'s (2011) article, "Does Algorithmic Trading Improve Liquidity?" and Andersen et al.'s (2003) article, "Micro Effects of Macro Announcements: Real-Time Price Discovery in Foreign Exchange" (refer to Table 2). The focus of the studies that received the most global citations was to provide early empirical evidence on the existence of PD in different market conditions and using different data models.

**Table 2.** Global citations.

| Authors | Year | Article Title | Journal | Total Citations |
|---|---|---|---|---|
| Hasbrouck, J. | 1995 | "One Security, Many Markets: Determining the Contributions to Price Discovery" | *The Journal of Finance* | 752 |
| Hendershott, T., Jones, C.M., Menkveld, A.J. | 2011 | "Does Algorithmic Trading Improve Liquidity?" | *The Journal of Finance* | 605 |
| Andersen, T.G., Bollerslev, T., Diebold, F.X., Vega, C. | 2003 | "Micro Effects of Macro Announcements: Real-Time Price Discovery in Foreign Exchange" | *American Economic Review* | 592 |
| Blanco, R., Brennan, S., Marsh, I.W. | 2005 | "An Empirical Analysis of the Dynamic Relation between Investment-Grade Bonds and Credit Default Swaps" | *The Journal of Finance* | 536 |
| Andersen, T.G., Bollerslev, T., Diebold, F.X., Claravegad | 2007 | "Real-time price discovery in global stock, bond and foreign exchange markets" | *Journal of International Economics* | 476 |
| Hasbrouck, J., Seppi, D.J. | 2001 | "Common factors in prices, order flows, and liquidity" | *Journal of Financial Economics* | 435 |
| Madhavan, A., Richardson, M., Roomans, M. | 1997 | "Why Do Security Prices Change? A Transaction-Level Analysis of NYSE Stocks" | *The Review of Financial Studies* | 426 |
| Brogaard, J., Hendershott, T., Riordan, R. | 2014 | "High-Frequency Trading and Price Discovery" | *The Review of Financial Studies* | 390 |
| Chakravarty, S., Gulen, H., Mayhew, S. | 2004 | "Informed Trading in Stock and Option Markets" | *The Journal of Finance* | 334 |
| Longstaff, F.A. | 2010 | "The subprime credit crisis and contagion in financial markets" | *Journal of Financial Economics* | 313 |

While discussing local citations, most citations were received by J. Hasbrouck's (1995) article, "One Security, Many Markets: Determining the Contributions to Price Discovery," followed by Baillie et al.'s (2002) article, "Price discovery and common factor models," and B. Lehmann's (2002) article, "Some desiderata for the measurement of price discovery across markets" (refer to Table 3). The focus of the studies cited the most in local citations was on providing different statistical and empirical models to test the PD process in different asset classes.

The study uses Bradford's law (Bradford 1934) in the price discovery discipline to understand the concentration and dispersion factors of publication patterns and the most productive nucleus. According to the law, a small nucleus of journals covers a larger proportion of publications in the discipline, whereas a larger nucleus of journals covers a smaller proportion (Alvarado 2016). Bradford's Law divides publications into three zones: Core, Zone 1, and Zone 2. Zone 1 and Zone 2 have n and n2 times the number of core journals, respectively. The Core, Zone 1, and Zone 2 ratios are 1:n:n2.

According to the findings of the analysis, a core nucleus of 15 (4.3% of total) journals caters to 478 (33.4%) publications. Zone 1 contains 55 (15.6% of total) journals covering 485 (33.9%) publications, while Zone 2 contains 282 (80.1% of total) journals covering only 468 (32.7%) publications (refer to Figure 3 and Table 4). This demonstrates a concentration of publications in the Core and an unequal distribution of publications across journals. Price discovery research is best served by a small number of journals, which publish a large proportion of the studies in the field.

**Table 3.** Local citations.

| Authors | Year | Title | Journal | Local Citations |
|---|---|---|---|---|
| Hasbrouck, J. | 1995 | "One Security, Many Markets: Determining the Contributions to Price Discovery" | *The Journal of Finance* | 363 |
| Baillie, R.T., Booth, G.G., Tse, Y., Zabotina, T. | 2002 | "Price discovery and common factor models" | *Journal of Financial Markets* | 191 |
| Lehmann, B. | 2002 | "Some desiderata for the measurement of price discovery across markets" | *Journal of Financial Markets* | 91 |
| Harris, F.H.D., Mcinish, T.H., Wood, R.A. | 2002 | "Security price adjustment across exchanges: an investigation of common factor components for Dow stocks" | *Journal of Financial Markets* | 90 |
| Hasbrouck, J. | 2003 | "Intraday Price Formation in U.S. Equity Index Markets" | *The Journal of Finance* | 89 |
| Yan, B., Zivot, E. | 2010 | "A structural analysis of price discovery measures" | *Journal of Financial Markets* | 82 |
| Chakravarty, S., Gulen, H., Mayhew, S. | 2004 | "Informed Trading in Stock and Option Markets" | *The Journal of Finance* | 78 |
| Jong, F.D. | 2002 | "Measures of contributions to price discovery: a comparison" | *Journal of Financial Markets* | 72 |
| Booth, G.G., So, R.W., Tse, Y. | 1999 | "Price discovery in the German equity index derivatives markets" | *The Journal of Futures Markets* | 71 |
| Eun, C.S., Sabherwal, S. | 2003 | "Cross-Border Listings and Price Discovery: Evidence from U.S.-Listed Canadian Stocks" | *The Journal of Finance* | 70 |

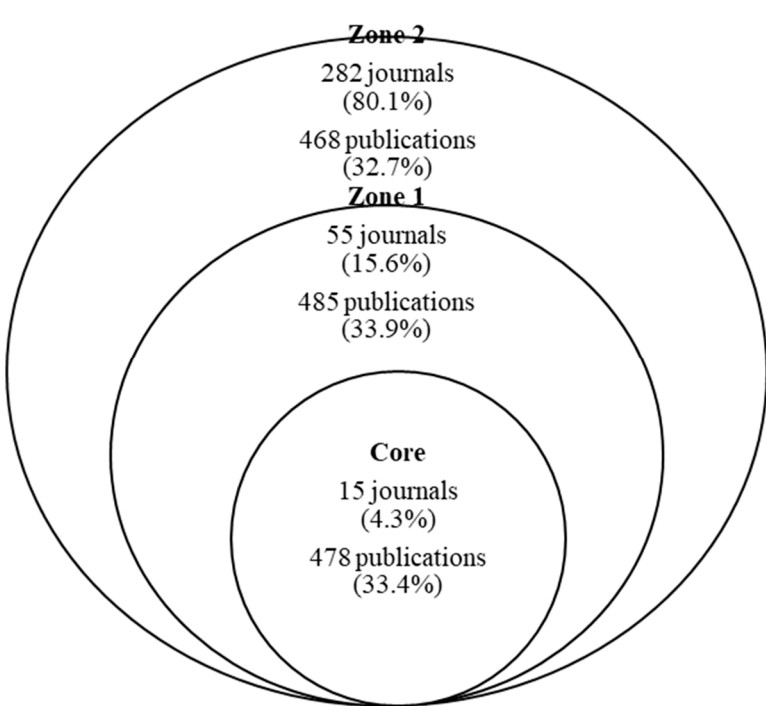

**Figure 3.** Dispersion of PD research in Bradford Rings.

**Table 4.** Publication dispersion zones of PD research under Bradford's Law.

| Nucleus | Journals | | Publications | | Ratio (1:*n*:*n*²) |
|---|---|---|---|---|---|
| | *N* | % | *n* | % | |
| Core | 15 | 4.3% | 478 | 33.4% | 1 |
| Zone 1 | 55 | 15.6% | 485 | 33.9% | 3.66 |
| Zone 2 | 282 | 80.1% | 468 | 32.7% | 13.44 |
| Total | 352 | 100.0% | 1431 | 100% | |

*3.3. Most Prominent Journals for Price Discovery Research (RQ3)*

The reputation of a journal within its field has an impact on the citation power of the publications in that journal. In the case of price discovery, scholars have published in journals across different disciplines (e.g., derivatives markets, banking and finance, financial markets, economics, and international money and finance). The journal-wise distribution of publications (refer to Table 5) indicates that the *Journal of Futures Markets* is the most highly preferred destination for scholars to publish their research on price discovery (101 publications, or nearly 7% of the total 1431 papers). *The Journal of Financial Markets* (44 publications, or about 3% of the total) and the *Journal of Banking and Finance* (37 publications, or about 2.6% of the total), along with the *Journal of Financial Economics* (15 publications), are also popular choices. Surprisingly, the top 10 journals only account for 22% of the total number of publications examined in this field. In terms of citations and identifying the most prominent source, the *Journal of Finance* comes at the top, followed by the *Journal of Futures Markets*. The *Journal of Futures Markets* ranks first in terms of impact as measured by the h-index, followed by the *Journal of Financial Markets*. The *Journal of Futures Markets* remains at the top of the table, followed by the *Journal of Financial Markets* when the impact is measured in terms of the g-index.

**Table 5.** Most relevant source and impact.

| Journal | NP | TC | h-Index | g-Index | m-Index | Start YP |
|---|---|---|---|---|---|---|
| *Journal of Futures Markets* | 101 | 2828 | 28 | 50 | 0.824 | 1989 |
| *Journal of Financial Markets* | 44 | 1972 | 22 | 44 | 0.917 | 1999 |
| *Journal of Banking and Finance* | 37 | 1269 | 17 | 35 | 0.586 | 1994 |
| *Journal of Financial Economics* | 25 | 1658 | 17 | 25 | 0.773 | 2001 |
| *Journal of Empirical Finance* | 24 | 578 | 13 | 24 | 0.619 | 2002 |
| *Journal of Finance* | 21 | 4039 | 20 | 21 | 0.769 | 1997 |
| *Energy Economics* | 20 | 438 | 13 | 20 | 0.406 | 1991 |
| *Journal of Financial and Quantitative Analysis* | 18 | 696 | 12 | 18 | 0.429 | 1995 |
| *Review of Financial Studies* | 14 | 1794 | 13 | 14 | 0.5 | 1997 |
| *Journal of International Money and Finance* | 14 | 456 | 12 | 14 | 0.4 | 1993 |

Notes: NP = Number of publications. TC = Total citations. Start YP = Starting year of publishing price discovery research. h-index = Publications with citations greater than/equal to h. g-index = g number of articles with g² citations. M-index = h-index per year since first publication.

*3.4. Most Prominent Authors Contributing to Price Discovery Research (RQ3)*

A total of 2461 authors have contributed to price discovery research. Out of this total, 1950 authors (79.2%) have published only one publication, while 308 authors (12.5%) have published two publications. A total of 31 authors (1.2%) have contributed more than five publications, whereas only 2 authors have contributed more than ten publications to the field (refer to Table 6). This shows that only a few scholars have made a significant contribution to price discovery research, which is still in the early stages.

**Table 6.** Author productivity of PD research.

| Documents Written | N. of Authors | Proportion of Authors |
|---|---|---|
| 1 | 1950 | 79.30% |
| 2 | 308 | 12.53% |
| 3 | 105 | 4.27% |
| 4 | 49 | 1.99% |
| 5 | 18 | 0.73% |
| 6 | 14 | 0.57% |
| 7 | 7 | 0.28% |
| 8 | 3 | 0.12% |
| 9 | 3 | 0.12% |
| 15 | 1 | 0.04% |
| 27 | 1 | 0.04% |

Out of the 2461 authors of price discovery research, the most productive author is Tse Y., who has contributed 27 publications since 1995, received 1197 citations, and has an h-index of 15 (Table 7). Following Tse Y. are Wang J. (15 publications) and Hendershott T. (9 publications), who have received 1524 and 1563 citations, respectively. Since no author in the field has an h-index above 20, it can be concluded that no author has a decisive influence on price discovery research, and the field can still be considered to be in the precursor stage of research.

**Table 7.** Author impact on PD research.

| Author | h-Index | g-Index | m-Index | TC | NP | Start YP |
|---|---|---|---|---|---|---|
| Tse Y | 15 | 27 | 0.536 | 1197 | 27 | 1995 |
| Wang J | 7 | 12 | 0.241 | 152 | 15 | 1994 |
| Hendershott T | 8 | 9 | 0.4 | 1563 | 9 | 2003 |
| Mcinish Th | 7 | 9 | 0.292 | 295 | 9 | 1999 |
| Chen Y L | 5 | 9 | 0.357 | 191 | 9 | 2009 |
| Frijns B | 6 | 8 | 0.429 | 127 | 8 | 2009 |
| Lien D | 6 | 8 | 0.3 | 174 | 8 | 2003 |
| Schwartz R A | 5 | 8 | 0.185 | 176 | 8 | 1996 |
| Corbet S | 7 | 7 | 1.4 | 181 | 7 | 2018 |
| Gau Y F | 5 | 7 | 0.357 | 181 | 7 | 2009 |

Notes: NP = Number of publications. TC = Total citations. Start YP = Starting year of publishing price discovery research. h-index = Publications with citations greater than/equals to h. g-index = g number of articles with $g^2$ citations. M-index = h-index per year since first publication.

Table 8 depicts the statistics of the top 10 contributing universities and institutions in price discovery research in the world, which contributed 194 publications (about 13.56% of total publications during this study period). The analysis of the affiliation reveals that the highest number of publications in the area of price discovery are from Australia, which contributes 66 articles (about 4.6%), followed by the United States, which also contributes 66 articles (about 4.6% of total publications during this study period). The university that has contributed the most to price discovery research is Auckland University of Technology in New Zealand, with 29 articles, followed by Deakin University and the University of Sydney in Australia, with 25 and 22 publications, respectively.

**Table 8.** Most relevant affiliations for PD research.

| Affiliation | Country | Articles |
|---|---|---|
| Auckland University of Technology | New Zealand | 29 |
| Deakin University | Australia | 25 |
| University of Sydney | Australia | 22 |
| Oklahoma State University | United States | 19 |
| University of Technology Sydney | Australia | 19 |
| University of Delhi | India | 18 |
| California State University | United States | 17 |
| International Islamic University Malaysia | Malaysia | 15 |
| Kansas State University | United States | 15 |
| University of Memphis | United States | 15 |

*3.5. Co-Authorship Network of Countries (RQ4)*

The co-authorship network of countries involved in price discovery research was analysed using VOS Viewer. The analysis reveals the major co-authorship networks within price discovery research. The USA is at the centre of the first major country collaboration network (purple), collaborating with China, Hong Kong, South Korea, and Canada (Figure 4). The second major country collaboration network (green) is driven by the United Kingdom, which collaborates with Sweden, Spain, Norway, Italy, Israel, and Greece. Australia is at the centre of the third major country collaboration network (blue), collaborating with Vietnam, Turkey, South Africa, and Latvia. India is at the centre of the fourth major country collaboration network (red), collaborating with France, Taiwan, Malaysia, the Netherlands, and Japan. Based on a close examination, the distribution of publications on price discovery research by nation reveals that the USA (n = 361, or 25.2% of all publications during this study period) has contributed the most publications, followed by China (n = 142, 9.9%), the United Kingdom (n = 90, 6.3%), Australia (n = 84, 5.9%), and India (n = 83, 5.8%) (refer to Table 9).

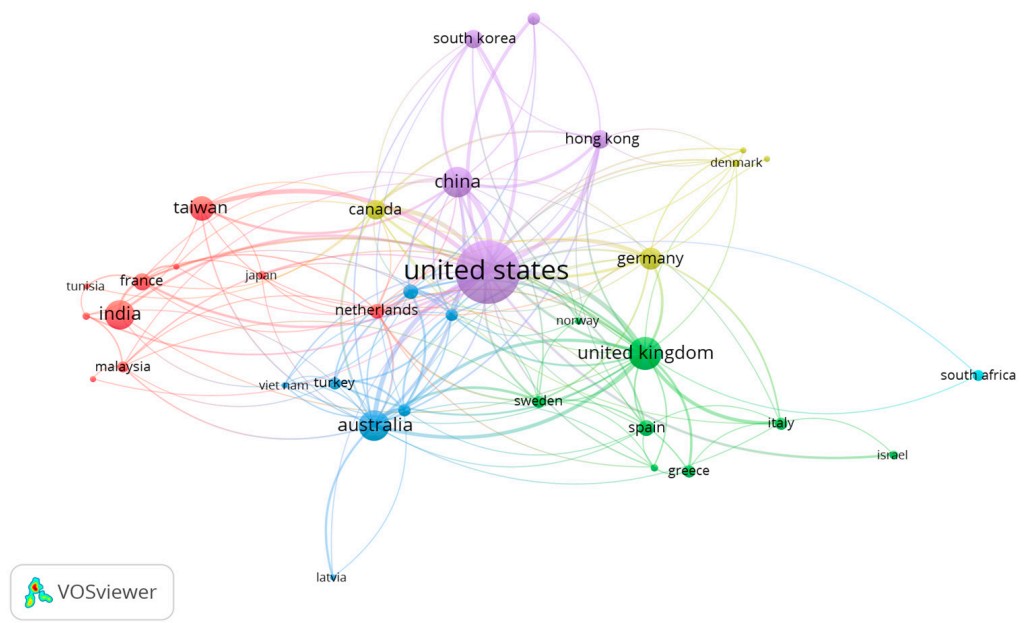

**Figure 4.** Co-authorship network among countries for PD research.

**Table 9.** Most relevant countries for PD research.

| Country | Articles |
|---------|----------|
| USA | 361 |
| China | 142 |
| United Kingdom | 90 |
| Australia | 84 |
| India | 83 |
| Germany | 41 |
| Korea | 33 |
| Canada | 30 |
| New Zealand | 29 |
| Spain | 24 |

*3.6. Author Collaboration in PD Research (RQ4)*

The following Table 10 shows the collaboration of the authors on PD research. The researchers Sifat I. M. and Mohamad A. have collaborated on six publications, followed by the co-authorship of five publications by Chen Y. L., Gau Y. F., Frijns B., and Tourani R. A.

**Table 10.** Bibliographic coupling of authors for PD research.

| Author 1 | Author 2 | Number of Articles Co-Authored |
|----------|----------|-------------------------------|
| Sifat I. M. | Mohamad A | 6 |
| Chen Y. L. | Gau Y. F. | 5 |
| Frijns B. | Tourani R. A. | 5 |
| Akyildirim E. | Corbet S. | 4 |
| Adrangi B. | Chatrath A. | 3 |
| Ante L. | Fiedler I. | 3 |
| Avino D. | Lazar E. | 3 |
| Beekes W. | Brown P. | 3 |
| Buckle M. | Chen J. | 3 |
| Dimpfl T. | Peter F. J. | 3 |

*3.7. Bibliographic Coupling of Authors' Affiliated Countries (RQ4)*

Figure 5 presents the bibliographic coupling of authors' affiliated countries, in which we set the coupling threshold of at least five publications as criteria. The United States is at the centre of this figure. Among the series of bibliographic couples, the coupling strengths of the USA and China, the USA and Australia, the USA and the United Kingdom, the USA and Hong Kong, the USA and Canada, the United Kingdom and Australia, and the United Kingdom and Italy are the strongest. These couplings are consistent with the findings in Table 8, which show that the country's most commonly affiliated authors are from the United States and Australia.

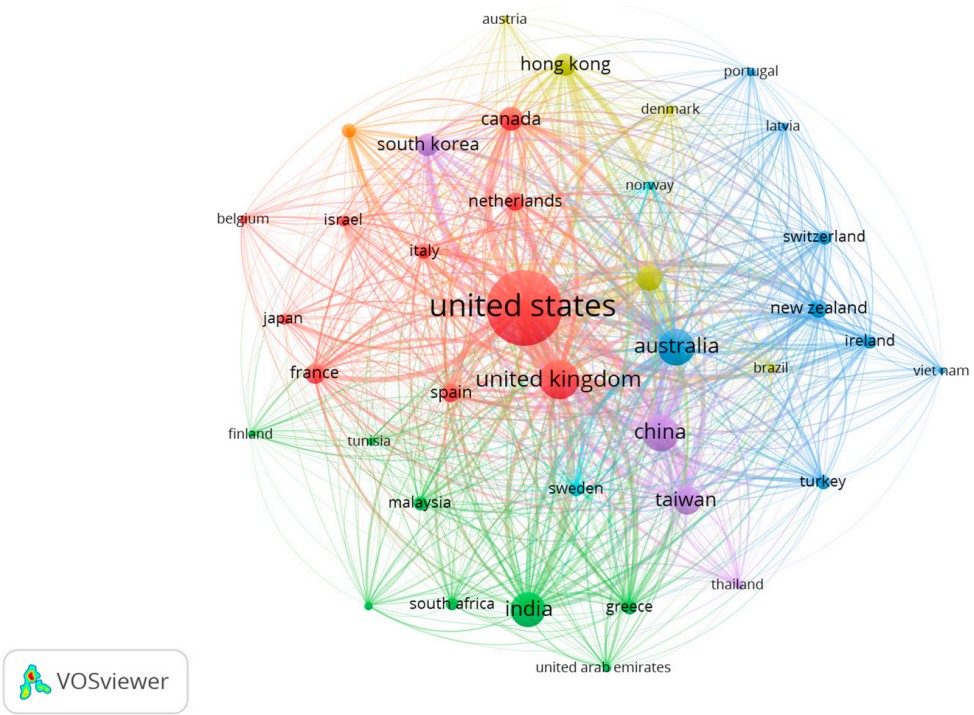

**Figure 5.** Bibliographic coupling of author–country for PD research.

### 3.8. Co-Occurrence of Author-Specified Keywords in Price Discovery Research Area (RQ5)

Figure 6 shows the frequency of author-specified keywords in the area of price discovery research, with the most common topics appearing at least five times in the publications between 1982 and 2021. The most frequent themes during 1982–2021 were price discovery, market efficiency, information asymmetry, market microstructure, volatility, futures, volatility spillover, etc.

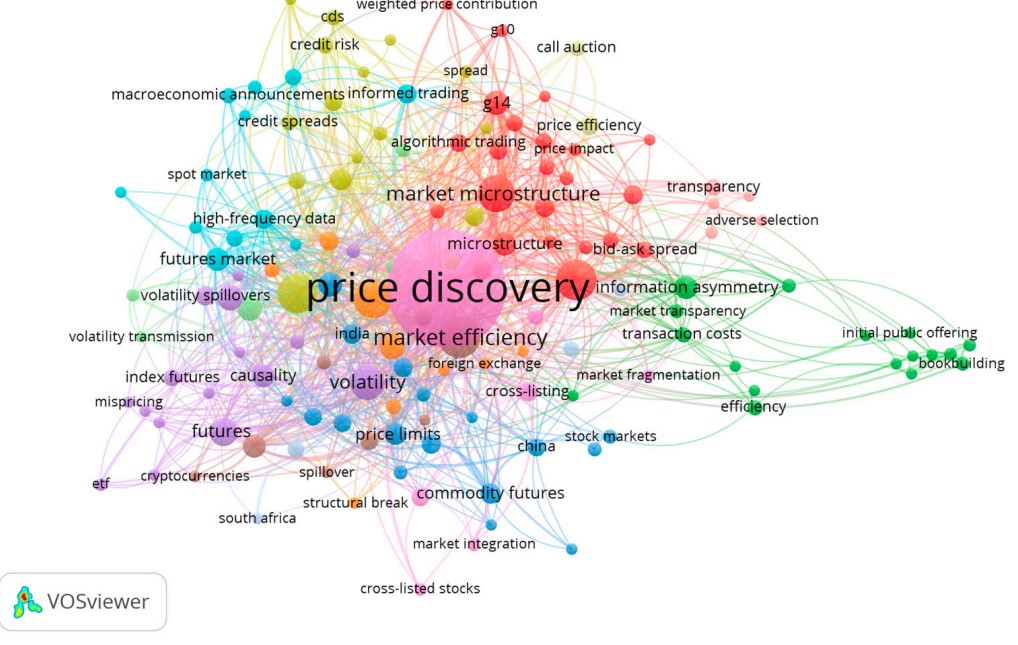

**Figure 6.** Co-occurrence of author-specified keywords for PD Research.

### 3.9. Co-Citation Network and Clustering of Articles (RQ5)

The literature also suggests that if two documents are being cited together, there is a possibility that these are falling into the same discipline of research (Hjørland 2013) or sharing similar contents (Small 1973). In the co-citation network of articles published in the field of PD, the data were presented as nodes and edges. The documents referred to were represented by nodes, while the links were shown by edges. The density of the edges, which is the link between nodes in one cluster that is on the higher side of the cluster compared to the nodes in other clusters, can be used as a parameter to divide nodes in a network into clusters (Mingers and Leydesdorff 2015). For a better understanding of the dispersion of the co-citation network, we classified it into three clusters. In each cluster, the top nine papers were selected by considering their page rank. The detailed analysis of each cluster is presented and summarised below (Figure 7).

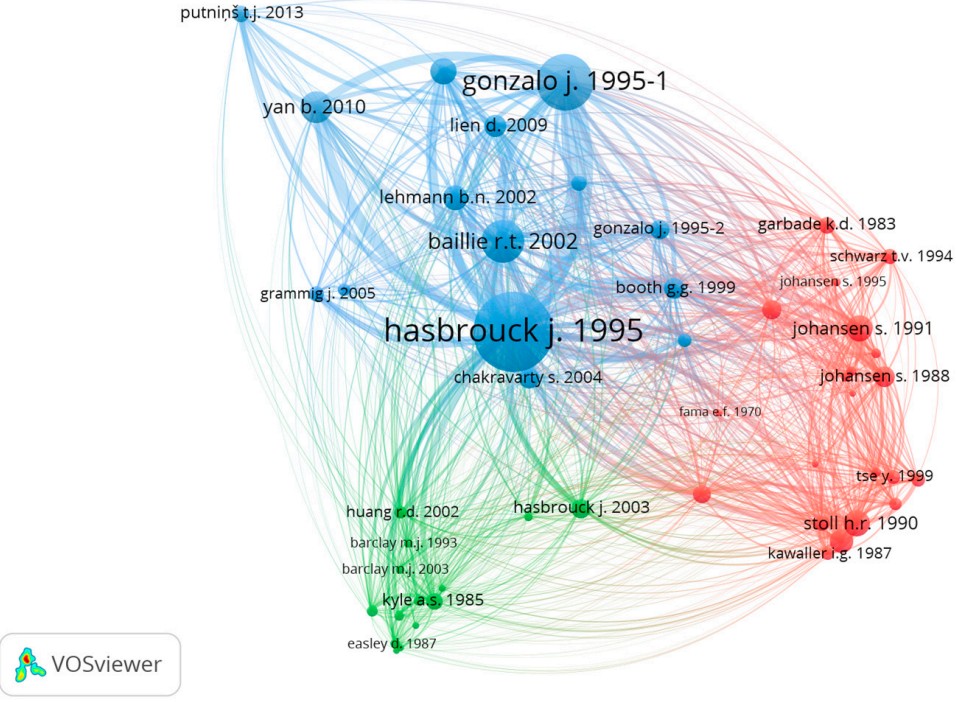

**Figure 7.** Clustering of Articles on PD using Co-citation Network.

**Cluster 1:** The study's first cluster serves as the foundation for the price discovery process. The cluster considers the study that laid down the theory to identify the price discovery process. The articles in this cluster provide a mechanism for evaluating the price discovery process under various market conditions. The foundation of the cluster was laid down by Garbade and Silber (1983), who explored whether the new information received in the market is first reflected in the futures of the security or in the spot prices of the underlying security in the market. The study by Hasbrouck (1995) dominates this cluster and justifies the logic of price discovery in closely-linked homogenous securities. This approach is considered an information-sharing approach in the existing literature on PD. The studies by Gonzalo and Granger (1995) and Baillie et al. (2002) suggested the use of the common method approach to test the price discovery process. Baillie et al. (2002) also tried to compare the factors proposed by Hasbrouck (1995) and Gonzalo and Granger (1995) and found that the common factors proposed by these two studies are related to each other. Later, Lien and Shrestha (2009) attempted to provide a modified information-sharing approach that did not rely on the upper and lower bounds of prices in the way that Hasbrouck's (1995) information-sharing approach did (refer to Table 11).

**Cluster 2:** According to co-citation analysis, cluster 2 is dominated by studies that suggest econometric tools and techniques to assess the price discovery process. The cluster considers studies by Johansen and Juselius (1990) and Johansen (1988), which advocate the use of a co-integrating equation to assess the long-run relationship between two non-stationary variables. Dickey and Fuller (1979) provide an econometric method for testing the stationarity of time series variables in another study. Before testing the long-term co-integrating relationship, it is important to assess whether the time series variables are stationary in nature. The same can be done after using the Dickey and Fuller (1979) unit root test method. Engle and Granger (1987) further suggested the use of the error-correction method after co-integration to see the long-run causality between the time series variables (refer to Table 11).

**Cluster 3:** Cluster 3 of the study considers the empirical studies that try to test the PD under different market conditions and asset classes. Cluster 3 is named "Price discovery under different market conditions and constraints" and is dominated by the study conducted by Kyle (1985). The study considers insider trading as a continuous action to test the information-sharing component in the price of the securities. Easley and O'Hara (1987) tried to explore the role of trade size and price of the underlying security to assess the PD, while Glosten and Milgrom (1985) used the bid–ask spread and transaction prices to assess the PD under "Specialized Markets that have heterogeneously informed traders." Amihud and Mendelson (1989) tried to assess the price discovery under different market microstructures in the Japanese market, and Hasbrouck (2003) tried to test the PD after considering the intraday prices in the US market. The other studies in the cluster also tried to explore the existence of PD in different markets and asset classes (refer to Table 11).

More specifically, in the case of equities, the existence of price discovery between the spot and future markets was assessed in the US market (Cornell and French 1983), in the Germany context (Booth et al. 1999), in the case of BRICS (Sharma et al. 2020), and in the Vietnam context (Rajput et al. 2012; Nhung et al. 2019). The researchers considered different data frequencies, sample periods, and empirical methodologies to assess the robustness of the existence of the PD process in these markers. In the US market, the study conducted by Cornell and French (1983) considered the daily closing data, while Kawaller et al. (1987) considered the intraday data to assess the same. Similar attempts were made in the case of commodities, currencies, and cryptocurrencies as asset classes. Considering the sample of eight commodities, Beck (1994) confirmed the existence of PD, while Yang et al. (2001) analysed the role of storage requirements for the commodities and assessed the existence of PD. The study discovered that PD exists for both types of commodities, including storable and non-storable commodities. In the case of other types of commodities such as energy (Shrestha 2014), metals, and agricultural commodities (Chinn and Coibion 2014; Dimpfl et al. 2017), the researchers have presented empirical evidence supporting the existence of PD. The researchers have made efforts to empirically test the price discovery in the currency market (Chen and Gau 2010; Osler et al. 2011; Rosenberg and Traub 2009; Kumar 2018; Sharma and Chotia 2019; Fassas et al. 2020; Akyildirim et al. 2020). Chen and Gau (2010) used the USD and Euro and Yen, Kumar (2018) considered the USD and INR, ZAR, and BRL, and Sharma and Chotia (2019) used the INR and USD, EURO, GBP, and JPY. In the case of cryptocurrency, mainly bitcoin (Karkkainen 2018; Sharma et al. 2022), Fassas et al. (2020) confirmed the existence of PD using the daily closing data, while the same was confirmed by Akyildirim et al. (2020) using the high-frequency data.

**Table 11.** Clustering of articles on PD research.

| Authors | Year | Title | Journal | Cluster | PageRank |
|---|---|---|---|---|---|
| **Cluster 1: Foundation of price discovery process** | | | | | |
| Hasbrouck, J. | 1995 | "One Security, Many Markets: Determining the Contributions to Price Discovery" | *The Journal of Finance* | 1 | 0.0850 |
| Gonzalo, J., Granger, C. | 1995 | "Estimation of Common Long-Memory Components in Cointegrated Systems" | *Journal of Business & Economic Statistics* | 1 | 0.0521 |
| Baillie, R.T., Booth, G.G., Tse, Y., Zabotina, T. | 2002 | "Price discovery and common factor models" | *Journal of Financial Markets* | 1 | 0.0431 |
| Yana, B., Zivot, E. | 2010 | "A structural analysis of price discovery measures" | *Journal of Financial Markets* | 1 | 0.0316 |
| Chakravarty, S., Gulen, H., Mayhew, S. | 2004 | "Informed Trading in Stock and Option Markets" | *The Journal of Finance* | 1 | 0.0218 |
| Lehmann, B. | 2002 | "Some desiderata for the measurement of price discovery across markets" | *Journal of Financial Markets* | 1 | 0.0273 |
| Jong, F.D. | 2002 | "Measures of contributions to price discovery: a comparison" | *Journal of Financial Markets* | 1 | 0.0279 |
| Garbade, K.D., Silber, W.L. | 1983 | "Price Movements and Price Discovery in Futures and Cash Markets" | *The Review of Economics And Statistics* | 1 | 0.0172 |
| Booth, G.G., So, R.W., Tse, Y. | 1999 | "Price Discovery in the German Equity Index Derivatives Markets" | *The Journal of Futures Markets* | 1 | 0.0233 |
| Lien, D., Shrestha, K. | 2009 | "A New Information Share Measure" | *The Journal of Futures Markets* | 1 | 0.0232 |
| **Cluster 2: Econometric tools and techniques to assess price discovery process** | | | | | |
| Johansen, S. | 1991 | "Estimation and Hypothesis Testing of Cointegration Vectors in Gaussian Vector Autoregressive Models" | *Econometrica* | 2 | 0.0333 |
| Johansen, S. | 1988 | "Statistical Analysis of Cointegration Vectors" | *Journal of Economic Dynamics And Control* | 2 | 0.0279 |
| Stoll, H.R., Whaley, R.E. | 1990 | "The Dynamics of Stock Index and Stock Index Futures Returns" | *The Journal of Financial And Quantitative Analysis* | 2 | 0.0345 |
| Engle, R.F., Granger, C.W.J. | 1987 | "Co-Integration and Error Correction: Representation, Estimation, and Testing" | *Econometrica* | 2 | 0.0223 |
| Chan, K. | 1992 | "A Further Analysis of the Lead–Lag Relationship Between the Cash Market and Stock Index Futures Market" | *The Review of Financial Studies* | 2 | 0.0300 |
| Fleming, J., Ostdiek, B., Whaley, R.E. | 1996 | "Trading Costs and The Relative Rates of Price Discovery in Stock, Futures, And Option Markets" | *The Journal of Futures Markets* | 2 | 0.0204 |
| Schwarz, T.V., Szakmary, A.C. | 1994 | "Price Discovery in Petroleum Markets: Arbitrage, Cointegration, and the Time Interval of Analysis" | *The Journal of Futures Markets* | 2 | 0.0171 |
| Tse, Y. | 1999 | "Price Discovery and Volatility Spillovers in the DJIA Index and Futures Markets" | *The Journal of Futures Markets* | 2 | 0.0180 |
| Dickey, D.A., Fuller, W.A. | 1979 | "Distribution of the Estimators for Autoregressive Time Series with a Unit Root" | *Journal of The American Statistical Association* | 2 | 0.0154 |
| Wahab, M., Lashgari, M. | 1993 | "Price Dynamics and Error Correction in Stock Index and Stock Index Futures Markets: A Cointegration Approach" | *The Journal of Futures Markets* | 2 | 0.0201 |
| **Cluster 3: Price discovery under different market conditions and constraints** | | | | | |
| Kyle, A.S. | 1985 | "Continuous Auctions and Insider Trading" | *Econometrica* | 3 | 0.0227 |
| Hasbrouck, J. | 1991 | "Measuring the Information Content of Stock Trades" | *The Journal of Finance* | 3 | 0.0146 |
| Hasbrouck, J., | 2003 | "Intraday Price Formation in U.S. Equity Index Markets" | *The Journal of Finance* | 3 | 0.0150 |
| Easley, D., O'Hara, M. | 1987 | "Price, Trade Size, And Information in Securities Markets" | *Journal of Financial Economics* | 3 | 0.0163 |
| Glosten, L.R., Milgrom, P.R. | 1985 | "Bid, Ask And Transaction Prices in A Specialist Market with Heterogeneously Informed Traders" | *Journal of Financial Economics* | 3 | 0.0141 |
| Admati, A. R., Pfleiderer, P. | 1988 | "A Theory of Intraday Patterns: Volume and Price Variability" | *The Review of Financial Studies* | 3 | 0.0113 |
| Huang, R.D., Stoll, H.R. | 2002 | "Tick Size, Bid-Ask Spreads, and Market Structure" | *The Journal of Financial and Quantitative Analysis* | 3 | 0.0124 |
| Amihud Y., Mendelson, H. | 1989 | "Market Microstructure and Price Discovery on The Tokyo Stock Exchange" | *Japan and The World Economy* | 3 | 0.0072 |
| Barclay M.J., Hendershott, T. | 2003 | "Price Discovery and Trading After Hours" | *The Review of Financial Studies* | 3 | 0.0117 |
| Barclay M.J., Warner, J.B. | 1993 | "Stealth trading and volatility. Which trades move prices?" | *Journal of Financial Economics* | 3 | 0.0113 |

## 4. Directions for Future Research (RQ6)

In view of the in-depth analysis of the literature published and the clustering of the existing themes of the studies published in the PD discipline, the directions for future research in the PD discipline are presented below.

1. Information quality plays a significant role in affecting the price discovery process in different asset classes. With the rising role of social media and other technology platforms, access to information has become very easy for end investors. This may also lead to a rise in the spread of both correct and misleading information to end investors on a real-time basis, and that may affect the price discovery discipline in different markets (Wu et al. 2022). This makes it important for the researchers to analyse the role of information quality through social media in affecting the price discovery mechanism across different asset classes and markets.

2. There is substantial evidence reported in the existing literature to assess the price discovery mechanism across different securities listed on different secondary markets, but the literature on the same in the case of over-the-counter (OTC) traded securities is limited. The volume traded in OTC markets is comparable to that traded on formal secondary exchanges (Lu and Zhan 2022). The researchers should try to explore the existence of price discovery in the OTC market.

3. The literature shows that some attempts have been made to assess the price discovery mechanism in the case of cryptocurrencies (an emerging asset class), but the same is limited to some of the more popular cryptocurrencies, including Bitcoin. A comprehensive analysis of the price discovery mechanism across different cryptocurrencies has yet to be presented in the literature, and thus becomes the focus of future research.

4. The major portion of the existing literature is focused on assessing the price discovery using the futures contracts as a measure of the information for future dates and seeing its impact on the price movement of the spot prices of the underlying securities. The options contracts are also made for a future date, and the option chain formed by the investors while investing in the options at different strike prices can also serve as a significant predictor for the price discovery mechanism across different asset classes. In the existing literature, not much focus has been given to options contracts to study the price discovery mechanism, and this is becoming a significant research gap for future researchers.

## 5. Conclusions

The analysis of the study presented above shows that the last decade (2012–2021) has witnessed the majority of the contribution (63%) in the price discovery discipline. Out of the total number of publications in the PD discipline, a small number of journals are preferred for price discovery research, publishing a large proportion of studies from the field. Some of these journals include *The Journal of Finance* and *The Journal of Futures Markets*. The focus of the most-cited studies using both global and location citation measures has been to provide empirical evidence on the existence of PD in different market conditions and using different data models. The study shows that only a few scholars have made a significant contribution to price discovery research, which is still in the early stages. Tse Y. and Wang J. are the most prominent authors, with 27 and 15 publications, respectively. The most frequent themes during 1982–2021 were price discovery, market efficiency, information asymmetry, market microstructure, volatility, futures, volatility spillover, etc.

From the cluster analysis, it is evident that the existing literature on PD can be divided into three clusters, where cluster 1 of the study provides the foundation of the price discovery process. Cluster 2 is dominated by the studies that suggest the econometric tools and techniques to assess the price discovery process, and cluster 3 of the study considers the empirical studies that try to test the PD under different market conditions and asset classes.

Considering this, the study suggests the directions for future research, including exploring the role of information quality through social media in affecting the price discovery mechanism across different asset classes and markets; conducting empirical studies to

explore the existence of the price discovery mechanism in the OTC market; conducting a comprehensive analysis of the price discovery mechanism across different cryptocurrencies; and considering options contracts to assess the price discovery mechanism across different asset classes and markets.

The study carries implications for researchers and professionals dealing in the field of risk management. The testing of price discovery mechanisms in different asset classes and markets allows the professional to assess whether the futures markets are acting efficiently compared to the spot market. The introduction of futures contracts is intended to provide instruments for hedging, and if the futures contracts are acting efficiently the overall risk management for different types of investors across different asset classes can be carried out efficiently. This also enables the professional to make accurate predictions about spot market movements and to trade in the market accordingly.

**Funding:** This research received no external funding.

**Data Availability Statement:** Not applicable.

**Acknowledgments:** We would acknowledge the support of editors and reviewers of MDPI for extending us necessary support for this research.

**Conflicts of Interest:** The authors declare no conflict of interest.

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
