# Peer review of "Research on Price Discovery in Financial Securities: Trends and Directions for Future Research"

_jrfm, doi:10.3390/jrfm16090416_

Round 1

Reviewer 1 Report

Please refer to the reviewing report for details. 

Please refer to the reviewing report for details. 

Author Response

Response to Reviewer’s Comments

Manuscript No. jrfm-2587036

We sincerely thank the Reviewer for the time and effort in reviewing our paper. After receiving the review reports of our manuscript, we did our best to address all the comments and suggestions. Please find below our point-by-point response.

Response to Reviewer 1

…………………………………………………………………………………………………

Comment: Research on Price Discovery in Financial Securities: Trends and Directions for Future Research The study uses the bibliometric analysis technique to do an in-depth analysis of the past research in the field of price discovery to find the research trends and directions for the future research in the area. Overall, I find that the paper idea is interesting and has potential contribution to the literature; The paper provides meaningful results and makes appropriate conclusion based on the results; Specifically, I was very impressed by three key clusters of the past price discovery literature identified by the study. I have one major comment and several minor suggestions (especially from the perspective of English expression) as follows:

Response: We thank the reviewer for expressing his interested in the article and appreciating the efforts. The major and minor suggestions received from the reviewer are address in subsequent sections. 

Comment: To start with, I would like to address the major comment. Specifically, I want to discuss the role of “section 2: Review of Past Study” in the manuscript. While it normally makes sense for a research paper to include “literature review” as the 2nd section, right after the introduction section, the inclusion of “literature review” as the 2nd section in this study sounds awkward. What is the purpose of reviewing past literature here? According to the 1st two sentence in the last paragraph of section 2 on page 3, the conclusion of the “literature review” is that “The review of studies presented above clearly shows that there are not many efforts made to study the in-depth literature of the PD discipline. No previous studies have attempted to analyze the literature published in the PD domain”. Stated differently, the purpose of the literature review is to argue for the “necessity of undergo the bibliometric analysis of the past studies in the paper”. The purpose coincides with the argument in the introduction section that “The discipline has been in existence for the past four decades, and to the best of our knowledge, no attempt has been made by the researchers to conduct an in-depth analysis of the existing literature on PD. Thus, it has become important to assess the past trends of the research publications on PD by conducting a comprehensive, in-depth analysis of the literature using the bibliometric method” (in the middle of the 3rd paragraph on page 2). If the purpose of the literature review is to argue for the necessity of doing bibliometric analysis of the past studies in the paper, there’s no need to include an additional section here. Thus, I suggest that the authors get rid of this section. In my opinion, there could be two potential purposes to do the literature review in this study: to highlight the importance of the study or to conclude for direction for the future research. Depending on the purposes of the literature review, the authors can potentially put the content of this section into two alternative areas:

- The first potential location to put some relevant literature review is the introduction section, somewhere at the 3rd paragraph of page 2. The authors can provide some additional empirical evidence to support the argument of the necessity of doing bibliometric analysis of the past studies in the paper.

Response: We thank the reviewer for suggestion. As per suggestion, we have removed the review of past studies section and the relevant content of the same is added in 3rd paragraph of page 2. 

Comment: - The second potential location to put some relevant literature review is at section 4.9, right after the description of literature within Cluster 3, as the additional literature review for the cluster. According to the abstract, “Cluster 3” contains “empirical studies to test the PD under different market conditions and asset classes”, which fits well with the description of the literature in section 2. After the additional literature review, the authors can naturally transit to the discussion of the directions for the future research in section 5.

Response: We thank the reviewer for suggestion. As per suggestion, we have removed the review of past studies section (section 2) and the relevant content of the same is added in discussion on cluster 3 presented in section 4.9 (revised section as 3.9 after removing section 2).

Comment: Next, I would like to address a few items from the perspective of English expression.

- I kept seeing the word of PD but could not find definition of PD in the manuscript. I think PD means “price discovery”. If this is the case, I suggest that the authors add the explanation “referred to as PD hereafter” in the first location where “price discovery” appears, probably in the abstract. - Please use more sentences in active voice within the manuscript, instead of sentences in passive voice.

Response: We thank the reviewer for suggestion. The suggested changes are incorporated in abstract of the study (as suggested).

Comment:- Please make sure that the description regarding an item is consistent in different locations of the manuscript. For example, the description of “Cluster 1” and “Cluster 2” is consistent in the abstract and the section 4.9. However, the description of “Cluster 3” is different. Specifically, the abstract describes the “Cluster 3” as “the empirical studies to test the PD under different market conditions and asset classes”, whereas section 4.9 names “cluster 3” as “Price discovery under different market conditions and constraints”. Even though the authors try to explain the two items in the 2nd paragraph on page 15, I think the two descriptions mean something different.

Response: We thank the reviewer for suggestion. The suggested changes are incorporated in abstract and cluster 3 of the study.

Comment:- The authors also need to pay attention to some English expressions, for example, the authors state “The researchers have made significant attempts to assess the existence of PD among different asset classes” in the 1st sentence of the 1st paragraph in section 2, and also state “.Significant attempts were made to assess the existence of the price discovery process in different asset classes” in the 1st sentence of the 2nd paragraph in section 2. What exactly are the difference between these two sentences? According to the two sentences, I think all the literature (reviewed in section 2) are related to the topic of “access the existence of price discovery in different asset classes”. If so, what is the point of dividing the discussion of the literature into two different paragraphs?

Response: We thank the reviewer for suggestion. As per previous suggestions, the section 2 of the study is removed, and the English expression used to represent the content in section 1 and cluster 3 are adjusted accordingly.

Comment:- Finally, I would suggest that the authors add some additional discussions to talk about the implications of the empirical findings in the conclusion section to highlight the topic of “risks”. For example, why is the study of the price discovery (of future contracts) important to corporation in term of risk management?

Response: We thank the reviewer for suggestion. The discussion on “why is the study of the price discovery (of future contracts) important to corporation in term of risk management?” is added in conclusion section.

Reviewer 2 Report

An interesting study with a large sample (1431 documents), provided significant results when applying the bibliometric method.

The introduction is well written, it is just necessary to present the study's aim.

The review of past studies presented a good view of the subject.

The methods is well explained

Related to the results, mainly Q1, Q2 and Q3, the results of a  literature review, moving beyond the simple act of tallying diverse documents or categorizing them into distinct sections, delve into a more nuanced approach. This entails a deeper exploration of each paper, transcending mere enumeration or superficial classification. We embark on a comprehensive journey through the content, where each document becomes a canvas for intricate analysis and thoughtful interpretation. Instead of brief annotations, we aim to unveil the essence, significance, and interconnections that permeate these papers, thereby fostering a richer understanding of their collective impact and contribution.

Author Response

Response to Reviewer’s Comments

Manuscript No. jrfm-2587036

We sincerely thank the Reviewer for the time and effort in reviewing our paper. After receiving the review reports of our manuscript, we did our best to address all the comments and suggestions. Please find below our point-by-point response.

Response to Reviewer 2

…………………………………………………………………………………………………

Comments: An interesting study with a large sample (1431 documents), provided significant results when applying the bibliometric method.

The introduction is well written, it is just necessary to present the study's aim.

The review of past studies presented a good view of the subject.

The methods is well explained

Related to the results, mainly Q1, Q2 and Q3, the results of a  literature review, moving beyond the simple act of tallying diverse documents or categorizing them into distinct sections, delve into a more nuanced approach. This entails a deeper exploration of each paper, transcending mere enumeration or superficial classification. We embark on a comprehensive journey through the content, where each document becomes a canvas for intricate analysis and thoughtful interpretation. Instead of brief annotations, we aim to unveil the essence, significance, and interconnections that permeate these papers, thereby fostering a richer understanding of their collective impact and contribution.

Response: We thank the reviewer for expressing his interested in the article and appreciating the efforts. The aims of the study are added in the introduction of the study.
